# The Lung Microbiome: A Central Mediator of Host Inflammation and Metabolism in Lung Cancer Patients?

**DOI:** 10.3390/cancers13010013

**Published:** 2020-12-22

**Authors:** Frank Weinberg, Robert P. Dickson, Deepak Nagrath, Nithya Ramnath

**Affiliations:** 1Department of Internal Medicine, University of Michigan Health System, Internal Medicine, Ann Arbor, MI 48109, USA; rodickso@umich.edu (R.P.D.); nithyar@umich.edu (N.R.); 2Department of Biomedical Engineering, University of Michigan, Ann Arbor, MI 48109, USA; dnagrath@umich.edu

**Keywords:** lung cancer, lung microbiome, metabolism, tumor microenvironment, inflammation, immunity

## Abstract

**Simple Summary:**

Lung cancer is the major cause of cancer related deaths in the world. New therapies have improved outcomes. Unfortunately, overall 5 year survival is ~20%. Therefore, better understanding of tumor biology and the microenvironment may lead to new therapeutic targets. The lung microbiome has recently emerged as a major mediator of host inflammation and pathogenesis. Understanding how the lung microbiota exerts its effects on lung cancer and the tumor microenvironment will allow for novel development of therapies.

**Abstract:**

Lung cancer is the leading cause of cancer-related death. Over the past 5–10 years lung cancer outcomes have significantly improved in part due to better treatment options including immunotherapy and molecularly targeted agents. Unfortunately, the majority of lung cancer patients do not enjoy durable responses to these new treatments. Seminal research demonstrated the importance of the gut microbiome in dictating responses to immunotherapy in melanoma patients. However, little is known regarding how other sites of microbiota in the human body affect tumorigenesis and treatment responses. The lungs were traditionally thought to be a sterile environment; however, recent research demonstrated that the lung contains its own dynamic microbiota that can influence disease and pathophysiology. Few studies have explored the role of the lung microbiome in lung cancer biology. In this review article, we discuss the links between the lung microbiota and cancer, with particular focus on immune responses, metabolism and strategies to target the lung microbiome for cancer prevention.

## 1. Introduction

Lung cancer is the leading cause of cancer-related death in the world. Even with the advancement in recent therapeutics and the development of immunotherapy, the 5-year overall survival in patients with non-small lung cancer (NSCLC) is <20%. Nonetheless, recent excitement for immunotherapy in lung cancer has centered around a minority of patients who develop durable, long-lasting responses to treatment, even in patients with advanced stage disease.

Moreover, a recent study demonstrated the importance of the gut microbiome in predicting response to immunotherapy in melanoma patients [1]. The lung represents an additional nidus of microbial colonization given its position at the interface between the internal and external host environments. Indeed, recent studies demonstrated the presence of a commensal lung microbiome in healthy subjects [2,3]. Further research has largely focused on understanding how the lung microbiome influences the pathophysiology of lung diseases; however, only a few studies have focused on understanding the influence of the lung microbiota on lung tumorigenesis.

Therefore, in this review we will summarize the current research surrounding investigation of the lung microbiome in lung diseases with a focus on lung cancer. Additionally, we will aim to elucidate the potential relationship between the lung microbiome, inflammation and metabolism. Finally, we will discuss potential therapeutic interventions that may arise out of further research investigating the role of the lung microbiome in lung cancer patients with a focus on how the lung microbiome may be exploited to improve responses and outcomes to immunotherapy.

## 2. The Lung Microbiome in Health

For many decades it was believed the lung represented a sterile environment. With the advent of culture-independent techniques and high-throughput sequencing of the 16S rRNA gene, a small highly conserved cistron of the bacterial genome, it became possible to identify the presence of bacterial communities at the genus or species-level phylogeny. In a landmark study in 2010, the authors published the first application of culture-independent techniques to identify microbiota present within the lungs of healthy patients as well as patients with asthma and chronic obstructive pulmonary disease (COPD) [3]. To date, there are over 30 published studies demonstrating the presence of bacterial communities in the lower respiratory tracts of healthy persons using modern culture-independent techniques [4]. In healthy lungs, the microbial composition is determined by the balance between immigration and elimination of bacteria [2,5]. Bacterial immigration occurs through inhalation of atmospheric air which can contain upwards of 10^4^–10^6^ bacterial cells per cubic meter [6] as well as microaspiration. Microaspiration is likely the primary source of immigration evidenced by overlap between the oral and lung microbiota [5]. It is well known that subclinical microaspirations occur in healthy persons [7,8]. Elimination of microbes from the lungs is a dynamic process that includes mechanical as well as immunological processes. Healthy airways contain ciliated epithelia that help to propel microbes proximally while the act of coughing represents an additional mechanism to expel microbes from the respiratory tract. It is well appreciated that the lung and airways contain specific innate and adaptive immunological defenses that allow for recognition and clearance of microbes. Further, the healthy lung is a heterogeneous environment with regional environmental variation that can influence bacterial composition. These variations include oxygen tension, pH, relative blood perfusion, relative alveolar ventilation, temperature, epithelial cell structure, deposition of inhaled particles and concentration and activity of inflammatory cells [9,10,11,12]. Dickson et al. proposed the adapted island model which postulates that in health the respiratory tract is one continuous ecosystem that is dynamic and varies with microbes originating from the single source of the upper respiratory tract and immigrating to the lower respiratory tract [9]. The number of microbial species at a specific site in the respiratory tract is a function of immigration and elimination factors. Dickson et al. validated this model, demonstrating that community richness (the number of species in a specific ecological community) decreases with increasing distance from the upper respiratory tract [2].

In general, studies indicate that a relatively uniform microbiota composes the healthy lung with the presence of certain dominant taxonomic groups [2]. The most abundant phyla are Bacteroidetes and Firmicutes with prominent genera including *Prevotella*, *Veillonella* and *Streptococcus* [13,14,15]. In general, the microbiota of the healthy lung most closely resembles that of the mouth, likely a result of microaspirations that occur during sleep when cough and laryngeal reflexes are diminished [5,16,17]. Within the lung of healthy persons there are considerable variations in environmental factors (described above); however, there is a relative lack of variation in the lung microbiome suggesting that the lung microbiota is in fact determined by the balance between immigration and elimination rather than specific growth factors. It has not been determined if the lung microbiome of healthy subjects varies geographically. One study assessed the lung microbiota from healthy subjects in eight US cities and found no evidence of geographic clustering [15].

## 3. The Lung Microbiome and Lung Cancer

Previous studies have linked the microbiota to various malignancies. More recently, studies performed by the Wargo group have correlated gut microbial composition to response to chemotherapy and immunotherapy in colon cancer models and advanced stage melanoma patients [1,18]. There have been a number of studies suggesting the gut microbiome plays an important role in various cancers, including lung cancer [19]. Further studies showed that the composition of the gut microbiota in lung cancer patients differed significantly from that of healthy control patients [20]. A systematic review of eight studies demonstrated that NSCLC patients treated with broad spectrum antibiotics prior to or during treatment with immune checkpoint inhibitors (ICI) had poorer clinical outcomes [21]. Indeed, this suggests that disruption of the gut microbiome with antibiotics may have an effect on response to ICI in NSCLC. It should be noted that antibiotic treatment would also have an effect on the lung microbiome as well. Additionally, many recent studies demonstrate the importance of the gut microbiome in modulating immune responses and changes in the gut microbiota alter immune responses and homeostasis in the respiratory system [22]. It is surmised that this cross-talk occurs through the production of metabolites, endotoxins and cytokines by the gut microbiota which travel to the respiratory mucosa through the bloodstream [23]. Some more recent studies suggest that this cross-talk is bidirectional, with the lung microbiota potentially mediating immunologic responses in the gut [24,25]. Given the abundance of data surrounding the gut-lung axis we chose to focus more fully on the effect of the lung microbiota directly upon lung tumorigenesis.

Characterization of the lung microbiome and its influences on lung cancer and treatment is in its infancy. Epidemiological studies have correlated *Mycobacterium tuberculosis* (TB) to lung cancer [26,27,28]. A systematic review of 41 studies demonstrated that a significantly increased lung cancer risk associated with pre-existing TB independent of tobacco use and exposure [27]. This suggests a direct link between TB and lung cancer and further suggests that the composition of the lung microbiome may have effects on tumorigenesis. In recent years, studies have demonstrated significant differences between the composition of the lung microbiome in lung cancer patients compared to healthy subjects or patients with alternative lung pathologies. In general, alpha diversity, the number (richness) and distribution (evenness) of taxa in a sample, is significantly higher in non-malignant lung tissues than tumor tissues while beta diversity, that is diversity in the microbial composition between different samples, is not significantly different between malignant and non-malignant tissues [29,30]. Other studies have determined specific taxa that are enriched in the airways of lung cancer patients. A study analyzing 216 lung aspirates collected from lung cancer patients demonstrated colonization of Gram negative bacteria such as *Haemophilus influenza*, *Enterobacter* and *Escherichia coli* [31]. It should be noted however, that the authors relied on culture-dependent assays for their analysis. In another study, sputum and oral samples were collected from female, never smoker, lung cancer patients (*n* = 8) as well as female, never smokers without cancer (*n* = 8). Samples taken from the lung cancer cohort had enrichment of *Granulicatella*, *Abiotrophia* and *Streptococcus* genera compared to the non-cancer cohort [32]. In addition, a study demonstrated enrichment of *Veillonella*, *Neisseria, Capnocytophaga* and *Selenomonas* in sputum collected from squamous cell and adenocarcinoma lung cancer patients (*n* = 20) as compared to non-cancer control subjects (*n* = 10) [33]. In a pilot study designed to identify potential bacterial biomarkers in lung cancer, the authors collected sputum from lung cancer patients (*n* = 4) and non-lung cancer subjects (*n* = 6) and found that seven specific bacterial species were present in all samples with significantly higher levels of enrichment of *Streptococcus viridans* in lung cancer samples [34]. More recently, bronchoalveolar lavage fluid (BALF) has been used to characterize the lung microbiome of the lower airways. Lee et al. collected BALF from 28 patients undergoing routine bronchoscopy for lung masses. Of the 28 patients, 20 were found to have lung cancer and 8 were diagnosed with benign diseases. Using culture-independent techniques the authors found increased relative abundance of the genera, *Veillonella* and *Megasphaera*, in lung cancer patients as well as Firmicutes and TM7 phyla as compared to patients with benign pathology [35]. Additionally, Tsay et al. collected airway brushings from patients undergoing routine diagnostic bronchoscopy for lung nodules from both the affected lung and contralateral unaffected lung. The authors report 39 subjects with lung cancer diagnoses, 36 subjects with non-cancer diagnoses and the inclusion of 10 healthy control subjects. They found that the lower airways of the lung cancer patients were enriched with oral tax (*Streptococcus* and *Veillonella*) as compared to non-cancer and healthy subjects. Interestingly, the uninvolved cancer airway samples shared many similar findings as seen in the involved cancer airway samples. It should be noted that this study included rigorous “control” specimens including buccal samples and bronchoscopy scope washes prior to procedures [36]. Together these findings support the presence of an altered lung microbiota in lung cancer patients and suggest enrichment with oral taxa likely a result of oral microaspirations (Figure 1).

Further analysis has involved understanding differences between microbial presence in cancerous lung versus noncancerous tissue. In one study, paired samples from tumor tissue and contralateral noncancerous sites were collected from lung cancer patients (*n* = 24) and healthy subjects (*n* = 18). Tissue from lung cancer patients had a significant decrease in microbial diversity as compared to healthy controls. *Streptococcus* and *Staphylococcus* were significantly more abundant in cancer patients compared to controls. The authors also found that the abundance of *Staphylococcus* and *Dialister* were highest in healthy subjects and declined in noncancerous tissues of lung cancer patients reaching lowest levels in tumor tissues [37]. This suggests that the lung microbiome may not only be important in early carcinogenesis but potentially plays a role in cancer progression. A recent study demonstrated that the lower respiratory tract microbiota predicts recurrence in patients with early-stage NSCLC after resection. Patnaik et al. collected pre-surgery BALF, lung tumor tissue and adjacent non-tumor tissue as well as saliva samples on 48 patients and found that BALF microbial signatures differed significantly between patients who had lung cancer recurrence within 32 months of surgery versus those who did not have a recurrence within 32 months. This was independent of age, sex, smoking status, tumor histology and tumor grade. Patients with recurrence were found to have a bacterial signature in BALF that was enriched with *Sphingomonas*, *Psychromonas* and *Serratia* while the abundance of *Cloacibacterium*, *Geobacillus* and *Brevibacterium* were reduced [38]. Another study looked at how the bacterial composition of the lung microbiota in lung cancer patients correlated to outcomes. In a pilot study, Peters et al. obtained paired lung tumor and distal normal tissue samples from the same area of the lung in 19 patients with NSCLC [39]. The authors demonstrated that patients with higher diversity and richness of their lung microbiota in unaffected lung tissue were associated with poorer disease-free (DFS) and recurrence-free (RFS) survival. In unaffected tissue greater abundance of family Koribacteraceae correlated with increased DFS and RFS while greater abundance of families Bacteroidaceae, Lachnospiraceae and Ruminococcaceae were associated with reduced DFS and RFS. Tumor tissue diversity and composition were not associated with RFS or DFS; however, tumor tissue had lower richness and diversity as compared to paired unaffected tissue [39]. To this end, the presence of certain bacteria has also been linked to varying stages of lung cancer in patients. Yu et al. collected 165 non-malignant lung tissue samples from lung cancer patients at various stages of disease. The authors found that advanced stage (IIIB, IV) patients had increased abundance of the *Thermus* genus while *Legionella* was higher in patients who developed metastatic disease [29]. These studies have several limitations including small sample sizes, collection of samples from different anatomical sites and lack of proper control. However, taken together these studies suggests a trend towards lung dysbiosis in association with lung cancer. These studies suggest not only the presence of a commensal microbiota present in lung cancer patients but the lung microbiota in lung cancer patients is altered and potentially transforms during the growth of a tumor, from initiation to progression.

Finally, cigarette smoking is still the major cause of lung cancer. Therefore, it is important to understand how tobacco smoke affects the airways and potentially impacts the lung microbiota. Multiple studies have demonstrated the significant impact tobacco smoke has on the upper airway microbiota, however, less is understood with regards to the lower airway microbiota. A study analyzing 64 BALFs from non-smokers and smokers did not find differences in bacterial composition of the lung between the two cohorts but did find differences in oral taxa [15]. Another study did not find changes in bacterial diversity after smoking cessation leading to the conclusion that smoking does not have a significant role in modifying the commensal lung microbiota [40]. However, this study was performed in patients who were asthmatics and used induced sputum for analysis. Furthermore, while cessation of smoking does lead to improved lung function and reduces COPD progression, it is unclear if smoking cessation has effects on lung remodeling and therefore, chronic exposure to tobacco smoke may remodel the lung microenvironment and subsequent commensal microbiome such that cessation of smoking may not have a great influence on the lung microbiota. Indeed, studies performed on humans and mice demonstrate that exposure to tobacco smoke alters the bacterial composition in the lower respiratory tract leading to altered and impaired local immune cell function [41,42,43,44]. Therefore, smoking may lead to dysbiosis in the commensal lung microbiota which allows for remodeling of the immune microenvironment within the lung allowing for either initiation, promotion or both, of tumorigenesis (Figure 1).

## 4. The Lung Microbiome and the Immune System

The lung sits at the interface between the outside environment and the internal host physiology and as such plays a fundamental role in innate and initial adaptive immunity in order to protect the lung from pathogenic insults. This is not unlike other tissues, such as the gut. Multiple studies in the gut have linked the gut microbiota to mucosal immunity and modulation of host immunity. Studies have demonstrated that the gut commensal microbiota regulates the innate immune system [45,46]. It is then expected that alterations in the commensal microbiota could potentially have significant effects on immune tone in the host. Indeed, studies have demonstrated that the Gammaproteobacteria class utilizes inflammatory byproducts to survive and propagate under low oxygen conditions. During conditions such as chronic inflammation, Gammaproteobacteria can outcompete bacteria that are unable to metabolize inflammatory byproducts for survival [47]. Gammaproteobacteria can use reactive nitrogen species, a byproduct of many inflammatory cells, as a terminal electron acceptor to support growth under conditions of inflammation [47,48,49]. Therefore, the conditions of the microenvironment have the potential to enrich for potentially pathogenic bacteria (i.e., Gammaproteobacteria) which in turn promote continued or chronic inflammation. As discussed previously, the microbiota composition of the lower airways in healthy lung is dominated by Bacteroidetes phylum that shifts towards Gammaproteobacteria (class which contains many lung-associated “pathogens”) in diseased airways. It is likely that in the lung the same mechanism of bacterial overgrowth occurs as it does in the gut with Gammaproteobacteria. This is evidenced in an number of studies in humans and mice which demonstrated that increased levels of Gammaproteobacteria in the lungs is associated with disease [4,50].

The lung has specialized alveolar macrophages (AM) and resident dendritic cells (DC) as well as other immune cells that monitor the lower airways for pathogenic insults. They are critical mediators of lung immune homeostasis ensuring that inflammatory and immune responses are activated in response to a pathogenic insult while dampening responses to harmless environmental stimuli. In general, the lung microenvironment is one of high immune tolerance. Both AMs and DCs stimulate the proliferation of regulatory T cells (Treg) and release prostaglandin E2 (PGE2), tumor growth factor-beta (TGF-B) and interleukin-10 (IL-10) which leads to a tolerogenic state [51,52]. Furthermore, it is now appreciated that commensal lung microbiota plays an important role in promoting immune tolerance through its effect on resident lung immune cells. One fundamental question is how commensal bacteria are recognized and tolerated by the lungs and immune microenvironment. Antigen presenting cells (APCs) in the lung, namely AMs and DCs, and lung epithelial cells express pattern recognition receptors (PRRs), such as Toll-like receptors (TLRs) as well as others, that recognize molecules of host and microbial origin. Activation of PRRs induces expression of immune related genes encoding for inflammatory cytokines, type I interferons and antimicrobial peptides and leads to the initiation of innate and adaptive immune responses [53,54]. PRR ligands are present on both commensal microbiota as well as pathogenic bacteria, however, commensal PRR ligands are thought be less agonistic than pathogenic ligands [55,56]. It is also appreciated that all innate immune cells are able to decipher between pro-inflammatory or danger signals produced by a pathogenic insult versus tolerogenic signals produced by non-damaged tissue, dietary components and commensal bacteria [57]. There are several mechanisms by which the commensal microbiota could allow for host tolerance. One way is through a mechanism in which commensal microbiota protect itself from immune detection by preventing the outgrowth and spread of potentially harmful microorganisms subsequently decreasing the risk of detection by the immune system [58]. It is also appreciated that the commensal microbiota is largely prevented from access to the host epithelium by mucous production and therefore commensal microbiota cannot stimulate epithelial cell PRRs [56].

On the other hand, pathogenic bacteria with virulence factors can easily breach the mucous layer and infiltrate the epithelium leading to an inflammatory response [59]. Together, this supports a model in which the host epithelia and immune microenvironment subscribe to an “ignorance is bliss” type model in which commensal microbiota is present but not detected. Furthermore, PRRs are not randomly distributed along mucosal surfaces and are strategically sequestered in areas where commensal bacteria are limited in their access [60,61]. To this point, the host and immune cells have developed mechanisms to tolerate commensal microbiota. Certain studies have shown that persistent PRR stimulation by microbiota derived signals preserve epithelial barrier integrity and TLR tolerance is achieved after persistent TLR stimulation [62,63,64]. Additionally, APCs continuously exposed to endotoxins leads to tolerogenic AMs and DCs [65,66]. Finally, PRR activation is also dependent on expression of dampening signals expressed by epithelial and immune cells, specificity for specific innate ligands and the cocktail of cytokines that shape a response to a specific PRR agonist [57,67].

As suggested, the lung microbiota is thought to play a role in immune tolerance by influencing APCs and Treg recruitment. Gollwitzer et al. found that a progressive shift from Gammaproteobacteria and Firmcutes towards Bacteroidetes in neonates induces increased expression of programmed-death ligand 1 (PD-L1) on dendritic cells which lead to necessary Treg development after birth [68]. Another study demonstrated that germ free (GF) mice sensitized and challenged with ovalbumin demonstrated increased airway reactivity and inflammation as compared with specific pathogen free (SPF) mice. Furthermore, when GF mice were reconstituted with commensal bacteria from the lungs of SPF mice there was decreased airway reactivity and inflammation. Additionally, GF mice were noted to have dysregulated DCs and AMs as compared to SPF mice, likely a result of the commensal lung microbiota’s ability to educate lung immune cells [69]. Further evidence of the lung microbiota’s ability to regulate host immune response is demonstrated in recent studies exploring the effect of viral insults and oral taxa on immune tone. Wang et al. examined the role of *Staphylococcus aureus*, an upper respiratory tract (URT) colonizer, and found that it was essential for augmenting resistance to lethal inflammatory responses to influenza viral challenges. SPF mice were found to have less *S. aureus* as compared to mice living in the natural environment and subsequently succumbed to death at higher rates due to induction of acute inflammation after influenza viral challenges. Furthermore, it was shown that *S. aureus* recruits monocytes into alveoli and induces polarization of AMs to an M2 phenotype leading to suppression of lethal inflammatory responses through release of anti-inflammatory molecules in response to influenza insult [70]. This study suggests that specific airway microbiota taxa can act as potential defenders against viral insults. In the age of Sars-Cov2 infection it would be interesting to analyze the lung microbiome from COVID-19 patients with hyper-inflammatory responses versus those with attenuated responses. Enrichment of specific bacterial taxa may be protective in those with attenuated inflammatory responses against Sars-Cov2 insult. Finally, Segal et al. demonstrated the presence of two distinct lower airway microbiota signatures in healthy subjects that correlated to differences in immune tone [71]. The authors found two distinct “pneumotypes”, one which is enriched with *Prevotella* and *Veillonella* (upper respiratory tract (URT) colonizers) termed supraglottic predominant taxa (SPT) and another population of persons termed background predominant taxa (BPT) characterized by low bacterial copy number whose BALF resembled background taxa environmental microbiota. Subjects with pneumotype SPT demonstrated increased numbers of lymphocytes in BALF, increased Th17 cells, IL-1α, IL-1β, fractalkine, IL-17, free fatty acids, inflammatory pathway mRNA and blunted TLR4 response. There was also decreased β-diversity and increased bacterial abundance [71]. Interestingly, the authors found that 45% of the healthy subjects are of pneumotype SPT which correlates to the percentage of microaspirators in the general population [7,8] again suggesting that microaspiration allows for the translocation of upper airway colonizers to the lower airway which affects lower airway inflammation and immune responses. This study suggests the importance of URT bacteria (*Prevotella* and *Veillonella*) in regulating the level of airway inflammation and Th17 immune activation in the lower airways. Additionally, co-culture experiments in which *Prevotella* and *Haemophilus* species were cultured with human DCs led to the observation that *Prevotella* could suppress *Haemophilus* induction of IL-12p70 in DCs leading to modulation of the immune response [72]. Another study demonstrated *Prevotella*-dominant airway microbiota was associated with development of more inflammatory prone macrophages [73]. This suggests that certain species of bacteria could actually modulate host immune response to other commensal or pathogenic bacteria. Given the microbiota’s effects on modulating lung inflammation, especially Th17 responses, it would not be surprising if these same upper respiratory commensals were playing a role in modulating tumor immunity. For example, a *Prevotella*-predominant microbiota in the lower airways could generate an immune microenvironment enriched with Th17 cell populations and IL-17 production which could support tumorigenesis. Indeed, IL-17A T cells have been linked to tumor invasion and metastasis in lung adenocarcinoma [74].

## 5. Immunity, Lung Microbiome and Tumorigenesis

Lung cancer is closely associated with chronic inflammation and the immune microenvironment is characterized by the accumulation of pro-inflammatory cytokines and pro-tumorigenic factors that can aid in tumorigenesis [75]. As discussed previously, commensal microbiota exist on mucosal surfaces exposed to the external environment. It is known that certain intestinal microbiotas promote inflammation and development of gastrointestinal cancers [76,77]. Recent studies have demonstrated the importance of commensal microbiotas in the gut as predictors of outcomes to immunotherapy in melanoma patients [1]. In lung cancer it should be noted that the clinical course is often characterized by frequent pulmonary infections and post-obstructive pneumonia which affects outcomes and suggests a potential link between the microbiome and lung cancer [78,79]. A recent study by Jin et al. attempted to link host-microbiota interaction to lung cancer development [80]. Using a Kras/p53 (KP) mouse model of lung adenocarcinoma the authors determined that GF mice were protected against lung cancer development as compared to SPF mice. Co-housing GF mice with SPF mice restored tumorigenicity. Furthermore, SPF mice treated at different stages of tumor growth with a cocktail of antibiotics had suppressed tumor growth in both early and advanced stages of disease. Additionally, characterization of the lower airway lung microbiota in tumor bearing KP mice demonstrated a commensal microbiome with increased bacterial burden, decreased bacterial diversity and enrichment with certain taxa such as *Herbaspirillum* and Sphingomonadaceae not observed in the lungs of non-tumor bearing mice [80]. These findings confirm that alterations in the commensal microbiota may play a significant role in modulating tumorigenesis. The authors went on to isolate and culture bacterial species from SPF mice and used them to inoculate KP mice shortly after tumor initiation and found a significant increase in disease development as compared to untreated mice. Moreover, the increased bacterial burden seen in these mice was correlated with increased expression of cytokines such as IL-1β and IL-23 within the tumor of tumor-bearing mice not seen in GF mice. Inhibition of AM and neutrophil induced IL-1β and IL-23 led to decreased tumor growth [80]. Together these findings suggest a direct link between the commensal microbiota and modulation of the micro-immune environment. Interestingly, the authors also found that increased γδ T cells were associated with SPF tumors as well as human lung adenocarcinoma samples and this increase was abrogated in GF mice. Tumor-associated γδ T cells were noted to have primarily expression of RORγt and IL-17A while γδ T cells in the peripheral lymph nodes and spleens of tumor bearing mice had significantly decreased IL-17 and RORγt levels as was also the case with GF mice. This supports the idea that the tumor microenvironment in which chronic inflammation is regulated through IL-17 producing cells, potentially allows for tumor development and growth. Depletion of the lung commensal microbiota in tumor bearing mice led to decreased γδ T cell abundance and IL-17A levels in BALF and serum. Additionally, co-housing GF mice with SPF mice restored γδ T cell abundance. Furthermore, it was found that tumor resident γδ T cells were not IFN-γ producers and their major functions involved IL-17A production and neutrophil infiltration and recruitment. This study is one of the first studies to directly link the lung commensal microbiota to tumorigenesis through modulation of the tumor immune microenvironment. Another study manipulated the lung commensal microbiota with aerosolized antibiotics in mice and found reduced numbers of Tregs, greater activation of immune effector cells and increased immunosurveillance in the lung which correlated to reduced growth of B16 melanoma lung metastases. This was associated with a decrease in *Streptococcus* and enrichment of Proteobacteria and Actinobacteria [81]. These studies suggest a direct link from microbiota to immune modulation, but how exactly the microbiota exerts its effects on immune cells in the microenvironment as well as the tumor is still unknown (Figure 2).

## 6. Immune Response, Lung Microbiome and Metabolism

One way the lung microbiota may exert it effects on immune cells and tumor are through microbial byproducts. Interestingly, Tsay et al. demonstrated that lung cancer patients had enrichment of the upper respiratory tract flora, especially *Prevotella*, *Veillonella* and *Streptococcus* [36]. When the authors cultured these bacteria individually it was found that the media in which *Veillonella* was cultured induced expression of ERK, PTEN, VEGFA and certain genes important to inflammasome function in A549 lung adenocarcinoma cells grown with the cultured media, suggesting that bacterial byproducts may be important for modulation of tumorigenic transcriptional pathways. While there are limited studies on lung microbiota-derived byproducts and the effects on host immune response as well as tumorigenesis, there have been a number of studies examining this relationship with regards to the gut microbiota. For example, bacterial derived acetaldehyde is known to be a carcinogen and deoxycholic acid (DCA) produced from the gut microbiota of obese individuals has been associated with hepatocellular carcinoma development [82,83]. More recently, bacterial-derived short chain fatty acids (SCFAs) have been shown to have anti-inflammatory effects in the gut which correlated to a decreased incidence of colon cancer [84]. *Clostridium* species present in the gut have been linked to the production of SCFAs and development of Treg cells [85,86,87]. Other members of the gut microbiota produce anti-inflammatory omega-3 polyunsaturated fatty acids as well as immune mediating tryptophan metabolites [88,89]. Therefore, understanding the composition of the microbiota will allow for a better understanding of the microenvironment and metabolic factors that could potentially be contributing to a pro-tumorigenic or anti-tumorigenic milieu.

Additionally, the gut microbiome has been demonstrated to have direct effects on host metabolism which represents another way the microbiota may influence the local microenvironment [90]. Gut microbiota perform essential functions in the metabolism of host bile acids, choline and phenols [91]. A recent study demonstrated that sphingolipids produced by the gut microbiota directly affect host lipid metabolism [92]. While the metabolic relationship between the lung microbiota and the host immune and tumor biology has not been explored in depth there are a few studies that attempt to link alterations in host metabolism to the lung microbiota. In particular, a study demonstrated that *Escherichia coli* and *S. aureus* may be able to increase proteases and protease inhibitors important to infection and immune response [93]. Furthermore, BALF isolated from HIV-infected patients, compared to healthy subjects, demonstrated differences in metabolite composition and this was associated with an altered lung microbiota. The presence of Caulobacteraceae, Staphylococcaceae and Nocardioidaceae contributed to alteration of metabolite levels in HIV patients [94]. Recently, a study demonstrated that propionate derived from the lung microbiota could induce cell cycle arrest and apoptosis in NSCLC cell lines suggesting bacterial derived metabolites can influence biologic pathways in cancer cells [95]. Bacterial metabolism can also modulate host immunity as evidenced by Segal et al. in which the authors found that healthy patients with enrichment of URT microbes such as *Prevotella*, *Veillonella* and *Streptococcus* had a relative increase in abundance of carbohydrate metabolism genes and decreased cellobiose and fucose-rhamnose indicative of active bacterial metabolism [71]. In addition, the study determined that enrichment of the lower airways with URT microbes led to increased bacterial metabolism that induced host cellular mucosal immunity of the Th17/neutrophilic phenotype and suppressed innate immunity [71]. Therefore, active bacterial metabolism may help to promote tumorigenesis through mediation of the Th17/neutrophilic response. Alternatively, it has been shown that host metabolic byproducts can have effects on local bacterial growth and composition. For instance, metabolites produced by immune cells, such as reactive nitrogen species, can promote growth of facultative anaerobes (i.e., Proteobacteria) on mucosal surfaces [50]. Together this research indicates an intricate crosstalk between the lung microbiota, the host immune system and tumor that may be mediated by bacterial metabolic byproducts or direct effects by the lung microbiota on host immune and tumor cell metabolic signaling (Figure 2).

## 7. Targeting the Lung Microbiota

Understanding the interplay between the lung microbiota and immune and tumor microenvironment will allow for potential new therapeutic strategies. These therapies could be designed to modulate commensal lung microbiota to induce a more tumor suppressive environment, target specific bacterial enzymes or byproducts important for either tumor growth, onco-immune responses, or both, modulate the resultant host biological effects induced by the lung microbiota or targeting the mechanism by which bacteria immigrate to the lung.

As discussed altered lung microbiota have the ability to attenuate immune responses in immune cells with certain species linked to the polarization of suppressive alveolar macrophages and decrease response to lipopolysaccharide leading to resistance to acute inflammation while other species have been linked to increased Th17 cell-mediated lung inflammation and decreased inflammation [70,71,96]. The ability to target specific bacterial species would allow for modulation of inflammatory responses potentially inducing a more anti-tumorigenic microenvironment. Nasal sprays and aerosolization could be exploited for this purpose. Indeed, multiple studies have demonstrated the efficacy of these interventions in delivering antibiotics, antibodies, cytokines, toll-like receptor agonists and bacterial cells [97,98,99,100,101]. In mice, it was found that the lung microbiota was altered after exposure to nasally instilled vancomycin while nasal administration of *Lactobacilli* was found to stimulate respiratory immunity and increase resistance towards viral infections [102,103]. In humans inhaled antibiotics have been used to treat critical lung infections while nasal instillation of *Streptococcus salivarius* has been shown to prevent acute otitis [101,104]. In a more recent study by Le Noci et al. it was demonstrated that commensal lung microbiota could be manipulated through antibiotic or probiotic aerosolization. The changes induced by these treatments led to reduction in immune suppression present in the lung microenvironment. Furthermore, the authors observed that decrease in bacterial flora induced through aerosolized antibiotic exposure reduced tumor implantation in the lung of mice and led to increased activation of NK and T cell effector cells. Using a model of melanoma in mice, the authors demonstrated a significant reduction in lung metastases by use of aerosolized antibiotics or certain Proteobacteria phylum species [81]. This suggests that modulation of lung microbiota in lung cancer patients could have profound effects on tumor growth and progression through remodeling of the immune microenvironment.

In addition, bacteriophages (BPs) could potentially be engineered against specific lung microbiota in order to shift the tumor and immune microenvironment in lung cancer patients towards a more anti-tumorigenic environment. BPs are viruses that can infect and kill bacteria but do not infect human or host cells. BPs are able to be genetically engineered to target specific bacterial species [105,106]. Numerous studies have demonstrated the beneficial effects of BPs for treatment of various infections. Oral administration of BPs was demonstrated to be effective in the prevention and treatment of cholera [107]. While topical use of BPs in skin infections such as diabetic foot ulcers as well as infections secondary to cutaneous burns have been shown to be effective in decreasing bacterial burden and treating infections [106,108]. Other studies have demonstrated the efficacy of BP treatment of chronic, refractory *Pseudomonas aeruginosa* otitis media or externa [109]. Aerosolized BPs have also been used to treat lung infections. Recently, aerosolized BPs were used to treat a patient with cystic fibrosis and chronic *Achromobacter* infection successfully while a small study compared administration of aerosolized BPs versus conventional antibiotic treatment for cystic fibrosis patients with *Pseudomonas aeruginosa* infection and found BPs reduced *Pseudomonas aeruginosa* concentration as well as need for additional antibiotics [110,111]. To the best of our knowledge there has not been a trial or study looking at the use of BPs in modulating the lung microbiome in lung cancer. This presents a potential tool for targeting specific altered commensal lung microbiota in lung cancer patients to potentially inhibit bacterial specific pro-tumorigenic effects with the added benefit of the absence of host side effects.

Targeting specific bacterial enzymes may also provide a way to modulate the lung microbiota and its effects on the tumor microenvironment. Bacteria, not unlike eukaryotic cells, rely on the biosynthesis of proteins, RNA, DNA as well as metabolites for their survival. In fact, microbial pathogenicity requires metabolic pathways in order to support bacterial growth and bacterial pathogens reprogram their metabolism to allow for survival [112]. Traditionally, targeting bacterial metabolic networks has not gained much therapeutic traction given significant overlap between bacterial and eukaryotic central metabolism however, there are multiple examples of exploitation of metabolic divergences between bacteria and eukaryotes for treatment of infections. For example, folate is an essential co-factor in the biosynthesis of nucleotides and is metabolized by dihydrofolate reductase (DHFR) leading to generation of nucleotides and certain amino acids. Both humans and bacteria encode a functional DHFR. Trimethoprim (TMP) selectively inhibits many microbial DHFRs leading to abrogation of essential metabolites and is used for the treatment of numerous infections. Additionally, bedaquiline was recently developed as an antitubercular drug targeting the F_0_F_1_ ATP synthase [113]. While conserved between mycobacteria and humans, the mycobacterial ATP synthase is 20,000 times more sensitive to bedaquiline than human mitochondrial ATP synthase [114]. By understanding the compositional make up at the species or genera level of the lung commensal microbiota in lung cancer additional studies could be undertaken examining the metabolic pathways present in altered commensal species in order to develop targeted therapeutics toward bacterial enzymes which would in turn affect the local tumor microenvironment.

As described, lung cancer tumorigenesis is characterized by an immune microenvironment enriched by Th17 cell responses along with expression of IL-17 as well as other cytokines. Additionally, lung commensal microbiota enriched with upper respiratory tract microbes (i.e., *Prevotella*, *Veillonella* and *Streptococcus*) leads to a Th17/neutrophilic phenotype within the lung microenvironment. Therefore, it can be surmised that lung microbiota enriched with URT microbes will support a pro-tumorigenic microenvironment through induction of Th17 responses. In the future, targeted treatments towards IL-17 combined with immunotherapy may yield increased responses to therapy in lung cancer patients. Indeed, Jin et al. found significantly decreased tumor growth, IL-1β and neutrophil infiltration when KP mice were treated with an IL-17A neutralizing antibody [80]. Currently, there are two FDA approved monoclonal antibodies targeting IL-17A, secukinumab and ixekizumab and one targeting the IL-17 receptor, brodalumab, approved for the treatment of psoriasis and psoriatic arthritis. To the best of our knowledge there are no current clinical trials investigating the use of these drugs in lung cancer patients. In the future, it may be reasonable to target downstream effects of the lung microbiota on inflammatory pathways to improve immunotherapy outcomes through modification of the immune microenvironment in lung cancer. Another cytokine, IL-6, has been linked to Th17 cell differentiation and therefore would present an alternative pathway to modulate the tumor immune microenvironment in lung cancer patients. Recently, a study demonstrated that relative abundance of Firmicutes phylum was positively correlated to alveolar IL-6 levels in patients with IPF [115]. Both *Streptococcus* and *Veillonella* genera are classified under the Firmicutes phylum and again this suggests that presence of URT microbes in the lower airway is associated with a pro-inflammatory phenotype. Furthermore, IL-6 is known to play an essential role in lung cancer by promoting COPD-like inflammation [116]. Therefore, targeting IL-6 in lung cancer patients may improve response to immunotherapy. Currently, the randomized Phase Ib/II Morpheus-Lung trial is investigating the safety and efficacy of the combination of tocilizumab, a monoclonal IL-6 antibody, with atezolizumab, a PD-1 monoclonal antibody (NCT03337698). Together, understanding how the lung microbiota influences the inflammatory/immune lung microenvironment in lung cancer patients could have significant impacts on the development of targeted therapies and use of immunomodulatory medications in combination with immunotherapy to improve outcomes (Figure 3).

In addition, as discussed, the pro-inflammatory effects of URT microbes present in the lower airways may have significant effects on tumor initiation and progression in the lung. It is now accepted that the enrichment of URT microbes in the lower airways is at least partly mechanical in nature as certain patients are pre-disposed to increased microaspiration. Therefore, interventional measures such as surgical procedures might play a role in preventing the enrichment and colonization of the lower respiratory tract with pro-inflammatory URT microbes. Alterations in the lung microbiome may also occur from increased bacterial burden through microaspiration due gastroesophageal reflux (Figure 1) [117]. Understanding the biological effects these gastrointestinal bacteria have within the lung and their role in modulation of the immune microenvironment would potentially provide a preventative strategy in which proton pump inhibitors could be used to prevent the immigration of pro-inflammatory/pro-tumorigenic bacteria from the gut into the lung.

Finally, understanding and characterizing the lung microbiome, especially in early stage lung cancer patients, may lead to the development of diagnostic lung cancer biomarkers. Recently, Poore et al. used The Cancer Genome Atlas (TCGA) to investigate the presence of unique microbial signatures in tissue and blood within and between 33 types of cancer from treatment-naïve patients. The authors were able to determine that blood-based microbial DNA signatures could discriminate within and between most types of cancer, including low-grade tumors, using commercial-grade cell-free tumor DNA platforms [118]. Together, this suggests that microbial DNA might be used in the future as a potential biomarker in lung cancer. Further studies would need to be performed in order to correlate microbial DNA present in the blood to the lung microbiota present in the lower airways.

## 8. Conclusions

It is well accepted that the lung harbors a dynamic microbiota that is influenced by a number of host and environmental factors. With the recent studies demonstrating the importance of the gut microbiota in predicting responses to immunotherapy in melanoma patients there has been renewed interest in understanding how the microbiome, including the lung commensal microbiota, modulates and remodels the tumor microenvironment. Further studies elucidating the role of the lung microbiome in metabolic and inflammatory regulation within the lungs of lung cancer patients may lead to the development of medical interventions designed to improve outcomes of treatments such as immunotherapy and aid in the evolution of preventative strategies for lung cancer patients.

## Figures and Tables

**Figure 1 cancers-13-00013-f001:**
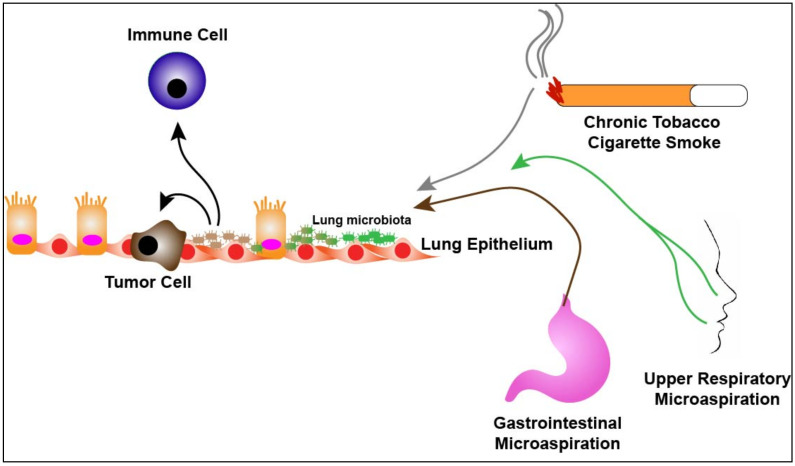
The lung microbiome in disease. Chronic exposure to either cigarette smoke, aerodigestive or both, microaspiration over time leads to dysbiosis of health-associated lung microbiota (green) towards microbiota enriched with upper respiratory tract bacterial species (brown). Respiratory dysbiosis potentially leads to promotion/enhancement of tumorigenesis as well as alteration of the lung immune microenvironment.

**Figure 2 cancers-13-00013-f002:**
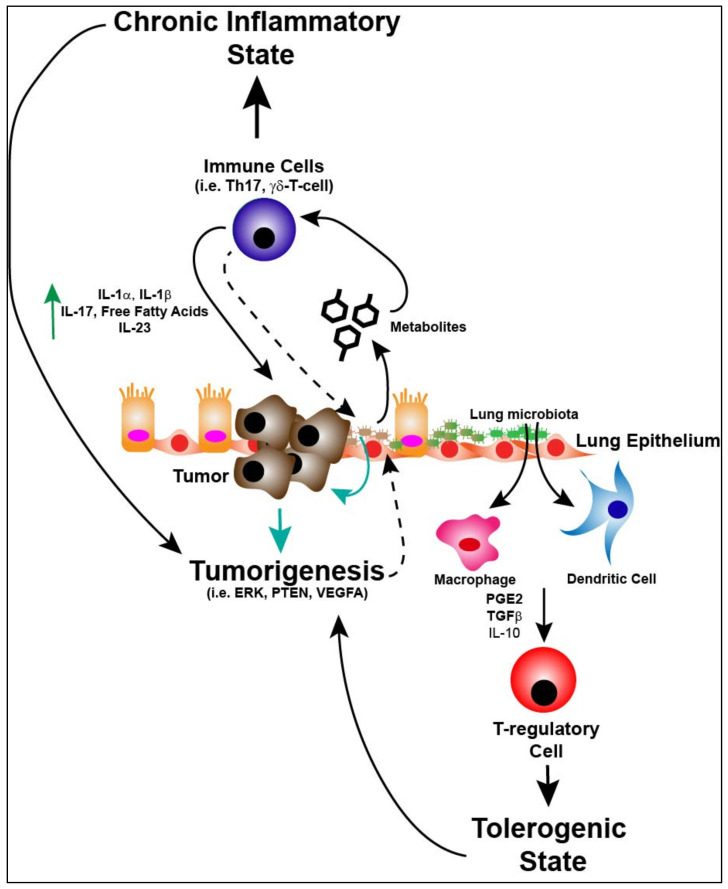
The lung microbiota as a central mediator of the tumor microenvironment and tumorigenesis. The microbiota of the lung can activate macrophages and dendritic cells in the microenvironment leading to T-reg cell induction through cytokines such as prostaglandin E_2_ (PGE2), transforming growth factor beta (TGFβ and interleukin-10 (IL-10) which leads to a “tolerogenic” state allowing for not only bacterial propagation and maintenance, but potentially tumor promotion. If respiratory dysbiosis occurs (i.e., from enrichment of upper respiratory tract bacterial species) this can lead to production of metabolic byproducts (i.e., deoxycholic acids, short chain fatty acids, omega 3 polyunsaturated fatty acids, tryptophan) which can induce a pro-inflammatory microenvironment and a “chronic” inflammatory state. Activation of immune cells (i.e., Th17 and γδ-T-cells) leads to production of cytokines (i.e., interleukin-1 alpha (IL-1α), interleukin-1 beta (IL-1β), interleukin-17 (IL-17), free fatty acids and interleukin-23 (IL-23) promoting tumorigenesis. Finally, dysbiosis of the lung microbiota can lead to direct effects on tumor intrinsic factors (i.e., extracellular signal-regulated kinase (ERK), phosphatase and tensin homolog (PTEN), vascular endothelial growth factor A (VEGFA). Additionally, the anatomic and immunologic consequences (dashed lines) of tumor biology change the local microenvironment for lung microbiota, perpetuating respiratory dysbiosis.

**Figure 3 cancers-13-00013-f003:**
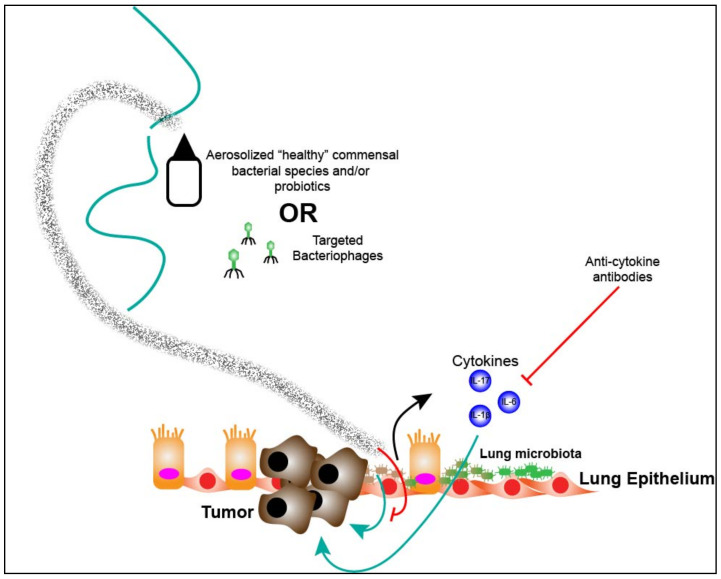
Targeting the lung microbiome. Aerosolization of “healthy” commensal respiratory bacteria, probiotics or bacteriophages targeted against dysbiotic bacteria in the lung could disrupt pathogenic lung dysbiosis in lung cancer patients and lead to anti-tumorigenic effects. Immunologic downstream effects of lung dysbiosis in lung cancer patients could lead to the use of specific anti-cytokine antibodies to modulate the inflammatory milieu promoting a more anti-tumorigenic tumor microenvironment within the lung.

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
