# Peer review of "The Lung Microbiome: A Central Mediator of Host Inflammation and Metabolism in Lung Cancer Patients?"

_cancers, 2020, doi:10.3390/cancers13010013_

Round 1
Reviewer 1 Report
Overview and general recommendation
This is a study on the role of lung microbiome in lung cancer, a subject already covered previously in excellent articles (PMID: 32061309, PMID: 31974656, PMID: 32953527, PMID: 32357415).
This review approaches this subject in a classical way but nevertheless original in its immunological aspects. However, several important points are missing from the review to make it a complete article suited for publication.
Major points
- Simple summary: the authors state that “Lung cancer is the major cause of cancer related deaths in the United States”. What is the situation in other countries (especially as the summary no longer specifies that it is the American epidemiology)? This article is intended to be read by non-Americans.
We have updated the paragraph to reflect WHO data which demonstrates that lung cancer is the leading cause of cancer related death world wide (1.76 million deaths in 2018).
- A quarter of the paragraph entitled "3. The Lung Microbiome in Disease" is not devoted to the lung cancer. This section deals superficially with a few diseases (but not exhaustively) other than lung cancer (asthma, COPD, IPF and infections), and does not contribute to the understanding of the main subject, lung cancer. One reason for addressing these diseases would be that there is a common core of pathophysiology with cancer, but this is not how the paragraph is approached. Therefore, this part is useless in the review; Numerous reviews on lung microbiome and various diseases have been published.
We have removed the section on the lung microbiome and other diseases and have renamed the section “3. The Lung Microbiome and Lung Cancer”.
- Lines 132-133: "Epidemiological studies have correlated Mycobacterium tuberculosis (TB) to lung cancer [34-36]. This suggests a link between the lung microbiome and lung cancer." The causal link between these 2 sentences needs to be explained. How does tuberculosis link to cancer?
We have provided further clarification and more details on the relationship between TB and lung cancer. We have added this on lines 158-161.
- The results presented one after one give the impression that a study corresponds to a series of results but that no general conclusions could be drawn. Could cancer typology (adenocarcinoma, squamous cell carcinoma, and large cell carcinoma) explain these differences, or only the fact that the presented studies are mainly case-control studies with a very little number or patients? These points have to be discussed (and eventually others issues, like the quality of the studies; Many of them described "contaminome" more than commensal lung microbiome).
We have addressed the many limitations in these small studies on page 7, line 227-229. We agree that the small sample sizes, different anatomical sites of collection as well as controls make each study hard to interpret in singularity. We would address the trend that all of the studies presented suggested lung dysbiosis is associated with lung cancer patients. Furthermore, the trend towards enrichment of Viellonella and Streptococcus in a number of the studies presented suggests a potential pattern of lung dysbiosis in lung cancer patients which warrants further investigation.
- The gut microbiome is an emerging link to lung homeostasis and disease (PMID: 33077630). In various places of the review, the gut microbiome is addressed but never in its links with the lung microbiome and pulmonary immunity, particularly in the crosstalk carried out viametabolic byproducts. These aspects should be presented in the article.
We agree that the gut-lung axis likely plays an important role in the modulation of the microbiome in both the lung and the gut. We also agree that this effect likely has an impact on immunity. We have added text introducing the concept of the role of the gut microbiome exerting effects on the respiratory tract in cancer. There have been a number of reviews focused on the gut-lung axis in lung cancer and our main objective in this review was to emphasize the importance of the lung microbiome and its direct effects on the respiratory tract in lung cancer patients rather than focusing significant attention on the gut microbiome. Please see page 4, lines 140-155.
- In figure 2, it is not understandable how the tolerogenic effect acts on tumor proliferation. This point has to be clarified in the figure.
We have updated Figure 2 so that it is clear how the tolerogenic effect induced by respiratory dysbiosis acts on tumor proliferation.
- In the role of the gut microbiome in melanoma, which is often taken as an example by the authors, the sequence in which the antibiotics are taken compared to the treatment by immunotherapy is very important in the success of the treatment. What about in lung cancer?
We have added text to address this point. Please see page 4, lines 123-147.
- The paragraph "7. Targeting the lung microbiota" would deserve to be summarized in the form of a figure.
We have created a figure to address this. Please see Figure 3 as well as figure legend.
Minor points
Several points have to be revised:
Line 132: Mycobacterium tuberculosis has to be written in italics.
We have made this correction.
Line 136: the number(richness)needs a space.
We have made this correction.
Line 141: gram negative has to be corrected in Gram negative (Gram comes from Mister Hans Christian Gram who discovered this staining).
We have made this correction.
Lines 191-193: family names have to be written in italics.
We have made these corrections.
Lines 257-258: « commensal PRR ligands are thought be less agonistic then pathogenic ligands » : "than" not "then"
We have made this correction
Lines 412 and bibliography: bacterial names have to be written in italics.
We have made these corrections.
Author Response
Dear Editor,
We would like to thank the reviewers for the time and thoughtful critique of our manuscript. Please see below for specific replies to each of their comments.
Overview and general recommendation
This is a study on the role of lung microbiome in lung cancer, a subject already covered previously in excellent articles (PMID: 32061309, PMID: 31974656, PMID: 32953527, PMID: 32357415).
This review approaches this subject in a classical way but nevertheless original in its immunological aspects. However, several important points are missing from the review to make it a complete article suited for publication.
Major points
- Simple summary: the authors state that “Lung cancer is the major cause of cancer related deaths in the United States”. What is the situation in other countries (especially as the summary no longer specifies that it is the American epidemiology)? This article is intended to be read by non-Americans.
We have updated the paragraph to reflect WHO data which demonstrates that lung cancer is the leading cause of cancer related death world wide (1.76 million deaths in 2018).
- A quarter of the paragraph entitled "3. The Lung Microbiome in Disease" is not devoted to the lung cancer. This section deals superficially with a few diseases (but not exhaustively) other than lung cancer (asthma, COPD, IPF and infections), and does not contribute to the understanding of the main subject, lung cancer. One reason for addressing these diseases would be that there is a common core of pathophysiology with cancer, but this is not how the paragraph is approached. Therefore, this part is useless in the review; Numerous reviews on lung microbiome and various diseases have been published.
We have removed the section on the lung microbiome and other diseases and have renamed the section “3. The Lung Microbiome and Lung Cancer”.
- Lines 132-133: "Epidemiological studies have correlated Mycobacterium tuberculosis (TB) to lung cancer [34-36]. This suggests a link between the lung microbiome and lung cancer." The causal link between these 2 sentences needs to be explained. How does tuberculosis link to cancer?
We have provided further clarification and more details on the relationship between TB and lung cancer. We have added this on lines 141-146.
- The results presented one after one give the impression that a study corresponds to a series of results but that no general conclusions could be drawn. Could cancer typology (adenocarcinoma, squamous cell carcinoma, and large cell carcinoma) explain these differences, or only the fact that the presented studies are mainly case-control studies with a very little number or patients? These points have to be discussed (and eventually others issues, like the quality of the studies; Many of them described "contaminome" more than commensal lung microbiome).
We have addressed the many limitations in these small studies on page 7, line 226-230. We agree that the small sample sizes, different anatomical sites of collection as well as controls make each study hard to interpret in singularity. We would address the trend that all of the studies presented suggested lung dysbiosis is associated with lung cancer patients. Furthermore, the trend towards enrichment of Viellonella and Streptococcus in a number of the studies presented suggests a potential pattern of lung dysbiosis in lung cancer patients which warrants further investigation.
- The gut microbiome is an emerging link to lung homeostasis and disease (PMID: 33077630). In various places of the review, the gut microbiome is addressed but never in its links with the lung microbiome and pulmonary immunity, particularly in the crosstalk carried out viametabolic byproducts. These aspects should be presented in the article.
We agree that the gut-lung axis likely plays an important role in the modulation of the microbiome in both the lung and the gut. We also agree that this effect likely has an impact on immunity. We have added text introducing the concept of the role of the gut microbiome exerting effects on the respiratory tract in cancer. There have been a number of reviews focused on the gut-lung axis in lung cancer and our main objective in this review was to emphasize the importance of the lung microbiome and its direct effects on the respiratory tract in lung cancer patients rather than focusing significant attention on the gut microbiome. Please see page 4, lines 140-152.
- In figure 2, it is not understandable how the tolerogenic effect acts on tumor proliferation. This point has to be clarified in the figure.
We have updated Figure 2 so that it is clear how the tolerogenic effect induced by respiratory dysbiosis acts on tumor proliferation.
- In the role of the gut microbiome in melanoma, which is often taken as an example by the authors, the sequence in which the antibiotics are taken compared to the treatment by immunotherapy is very important in the success of the treatment. What about in lung cancer?
We have added text to address this point. Please see page 4, lines 141-144.
- The paragraph "7. Targeting the lung microbiota" would deserve to be summarized in the form of a figure.
We have created a figure to address this. Please see Figure 3 as well as figure legend.
Minor points
Several points have to be revised:
Line 132: Mycobacterium tuberculosis has to be written in italics.
We have made this correction.
Line 136: the number(richness)needs a space.
We have made this correction.
Line 141: gram negative has to be corrected in Gram negative (Gram comes from Mister Hans Christian Gram who discovered this staining).
We have made this correction.
Lines 191-193: family names have to be written in italics.
We have made these corrections.
Lines 257-258: « commensal PRR ligands are thought be less agonistic then pathogenic ligands » : "than" not "then"
We have made this correction
Lines 412 and bibliography: bacterial names have to be written in italics.
We have made these corrections.
Reviewer 2 Report
In the manuscript (cancers-975360) entitled “The lung microbiome: a central mediator of host inflammation and metabolism in lung cancer patients?” Weinberg et al. review the current literature on the role played by lung microbiome on lung cancer development and progression. Whereas a link between gut microbiota composition and colon cancer is well documented by various publications studies examining how the lung microbiome can shape the metabolic and immune response of the lung and thus cancer development and progression are still limited. Therefore, the major merit of the present review is to point out the importance of studying lung microbiome to better understand lung cancer biology, which is pivotal to develop more effective therapeutic interventions.
Author Response
We appreciate Reviewer 2's comments and time and commitment to reviewing our manuscript.
Round 2
Reviewer 1 Report
The authors responded well to the comments and improved their manuscript, especially with the figures.